REGISTERED REPORT PROTOCOL

# Prevalence and determinants of undernutrition among adolescents in India: A protocol for systematic review and meta-analysis

**Jayashree Parida[1], Lopamudra Jena Samanta[1], Jagatdarshi Badamali[1], Prasant Kumar Singh[2], Prasanna Kumar Patra[3], Bijay Kumar Mishra[1], Sanghamitra Pati[1], Harpreet Kaur[4], Subhendu Kumar Acharya[1]***

**1** Indian Council of Medical Research -Regional Medical Research Centre, Bhubaneswar, Odisha, India, **2** Indian Council of Medical Research–National Institute of Cancer Prevention and Research, New Delhi, India, **3** Department of Anthropology, Utkal University, Bhubaneswar, Odisha, India, **4** Division of Tribal Health, Indian Council of Medical Research, New Delhi, India

* a.subhendu@gmail.com

## Abstract

### Background

Undernutrition is one of the serious health problems among adolescents in India where 253 million adolescents are in the age group of 10–19 years. Since adolescents represent the next generation of adults, it is important to understand the nutritional status of adolescents. Even though several studies have been carried out in different states in India on adolescent undernutrition (stunting, wasting /underweight), there is no study or review that estimated the national pooled prevalence of adolescent undernutrition and its determinants. Therefore, this review aims to determine the pooled prevalence and determinants of undernutrition (stunting, underweight/wasting) among Indian adolescents.

### Methods

A systematic review of eligible articles will be conducted using preferred reporting items for systematic reviews and meta-analysis (PRISMA) guidelines. A comprehensive searching of the literature will be made in Pub Med, EMBASE, SCOPUS, Google, Google Scholar, and Cochrane databases. The quality of the articles included in the review will be evaluated using the Newcastle-Ottawa Scale (NOS) for observational studies in meta-analyses. The pooled prevalence and odds ratio of the associated risk factors or determinants with their 95% confidence interval will be computed using STATA version 16 software. The existence of heterogeneity among studies will be assessed by computing p-values of Higgins's $I^2$ test statistics and Cochran's Q-statistics based on chi-square with a 5% level of significance among reported prevalence. Sensitivity analysis and subgroup analysis will be conducted based on study quality to investigate the possible sources of heterogeneity. Publication bias will be assessed through visual examination of funnel plots and objectively by Egger's regression test. This review protocol has been registered at PROSPERO (**CRD42021286814**).

**Data Availability Statement:** All relevant data are within the paper and its Supporting Information files.

**Funding:** The author(s) received no specific funding for this work.

**Competing interests:** The authors have declared that no competing interests exist.

## Discussion

By collecting and summarizing information on adolescent undernutrition can be a step towards a better understanding of the prevalence of nutritional status of Indian adolescents and how the associated factors influence the prevalence of undernutrition. This review will provide directions for further research and healthcare practitioners. This summarized finding at the national level will provide impetus to build nutritional strategies and proper healthcare services to fight against undernutrition among the most ignored population.

## Introduction

The problem of undernutrition is a global concern in recent years and it is increasing at a wider rate in developing countries like India [1]. The main reason for focusing the analysis on undernutrition is that undernutrition continues as an economic 'macro' issue related to food entitlements, poverty, and the socio-economic structure of societies as well as a primary cause of disease susceptibility, morbidity and mortality throughout the Third World [2, 3]. In India particularly, despite lots of initiatives and development, it continues to be high among children and adolescents in both the rural and urban areas [4–7]. Lots of interventions have been taken by the government to reach out to adolescents and increase their access to public health and nutrition services, but the issue of undernutrition among adolescents has not received adequate attention in policy discourse. The study on the nutritional status of adolescents is important because they are the future generation adults and it will help to plan strategies for combating macro and micro nutritional deficiencies of future citizens [8]. The systematic review of undernutrition at national and local levels studies have focused predominantly on children under 5 years of age. Focusing on adolescents, the age group, with the highest growth rapidity after infancy is lacking.

According to the WHO, adolescence is defined as the age group of 10–19 years [9]. The adolescent period is a very important phase in the life span of an individual. It is defined as the transition period from childhood to adulthood and is characterized by exceptionally rapid growth. During this stage, the adolescents experience major biological and psychological changes often shaped by socio-cultural factors [10] which need major nutrition. There are two indicators for measuring undernutrition among adolescents: the low BMI (Body Mass Index) for age i.e. $< -2$ standard deviation (SD) and stunting, the low height for age i.e. $<-2$SD. Asia has more than half of the world's adolescents while according to the Census 2011, 20% of India's population are adolescents. India comprises 18.02% of the adolescent population of the World total population [11]. Though the prevalence of thinness in boys and girls has decreased from 58.1% and 46.8% in NFHS-3 to 45% and 42% in NFHS-4 respectively, still it is a challenging issue for a developing country like India [10]. Low literacy levels, poor socioeconomic status, poor environmental hygiene, poor dietary habits aggravate undernutrition among Indian adolescents [12–15]. To improve and break the intergenerational cycle of adolescent undernutrition, there is a need for a strong evidence-based study and this review-based study will fulfill this aspect by providing the estimates of the prevalence of undernutrition. Several studies evaluated adolescent undernutrition and its related risk factors in India. Moreover, no studies or reviews estimated the pooled prevalence of adolescent undernutrition and its determinants. Therefore, this systematic review and meta-analysis will provide the estimation of the pooled prevalence and determinants of undernutrition among adolescents in India which can help the policymakers and public health professionals for decision making.

### Research questions

i. What is the pooled prevalence of undernutrition (stunting, underweight/wasting) among adolescents in India?

ii. What are the major determinants of undernutrition (stunting, underweight/wasting) among Indian adolescents?

### Objectives

i. To estimate the pooled prevalence of undernutrition (stunting, underweight/wasting) among adolescents in India

ii. To identify the determinants of undernutrition (stunting, underweight/wasting) among Indian adolescents.

## Materials and methods

### Systematic review registration and reporting of the findings

The protocol for this systematic review and meta-analysis has been prepared according to the Preferred Reporting Items for Systematic review and Meta-analyses (PRISMA-P 2009) [16], the Meta-analysis of Observational Studies in Epidemiology [17], and the PRISMA-P 2015 checklist [18]. The PRISMA-P checklist is given in S1 File. This review protocol has been registered at the international prospective register of systematic review and meta-analysis (**PROSPERO: CRD42021286814**).

### Study design and search strategy

The study is a systematic review, aims to determine the pooled prevalence of undernutrition among Indian adolescents of aged 10 to 19 years and also to identify the associated determinants of undernutrition among Indian adolescents.

Before the initiation of the review process, databases were searched to check whether the same systematic review and meta-analysis is present or not to avoid duplicacy. The Preferred Reporting Items for Systematic Review and Meta-Analysis will be used for the preparation of this systematic review and Meta-analysis. For this review, studies published between 2000 and 2021 in PubMed, EMBASE, SCOPUS, Google, Google Scholar, and Cochrane databases will be systematically searched. The articles will be searched separately by the reviewers (J.P., L.J. and J. B.) using the following keywords: "adolescents", "teenagers", "young adults", "undernutrition", "nutritional deficiency", "micronutrient deficiency", "nutritional status", "underweight", "stunting", "wasting" and "India" and also in combination using the Boolean operators like "OR" or "AND". The searching strategy to be used for PubMed online database is given in S2 File. Also, the reference lists of the included articles will be searched to find out the eligible articles. Grey literature will be retrieved using Google and Google Scholar. The procedures of screening and selection of eligible studies will be presented by using the PRISMA flow diagram.

### Eligibility criteria

**Inclusion criteria**.

i. Population: adolescents (10–19 years-old).

ii. Exposure: Determinants of undernutrition.

iii. Comparator: The prevalence of undernutrition (stunting and wasting/underweight) in rural and urban areas, the prevalence of undernutrition among boys and girls, nutritional status among early adolescents, and late adolescents.

iv. Type of studies: All cross-sectional, cohort, and case-control observational studies indicating the prevalence of undernutrition among adolescents.

v. Studies conducted in India only.

vi. Anthropometric indicators: stunting, wasting or thinness or underweight.

vii. Growth reference: WHO growth standards.

viii. All articles published only in the English language.

**Exclusion criteria.** Studies will be excluded if they include:

i. Only overnutrition indicators (obesity and overweight).

ii. Children under 10 years and adults.

iii. Studies on micronutrient deficiency.

iv. Adolescent pregnant women.

v. Studies conducted among the special population, such as studies done among adolescents living with anemia or other diseases like HIV/AIDS.

vi. As smaller studies increase the risk of bias, studies with a sample size of less than 100 will be excluded.

### PECO search guide

**Population.** Adolescents (10–19 years-old).

**Exposure.** Determinants of undernutrition. The determinants are the exposures that increase the prevalence of undernutrition among adolescents. These risk factors are the educational and occupational status of parents, household income, age, sex, food security, and dietary habits.

**Comparator.** The reference group of each determinant in each study. The prevalence of undernutrition (stunting and underweight/wasting) in rural and urban areas, prevalence of undernutrition among boys and girls, nutritional status among early and late adolescents.

**Outcome.** The prevalence of undernutrition (stunting and underweight/wasting) and its associated factors or determinants among Indian adolescents.

### Outcome measurement

Anthropometric indicators namely stunting, underweight, wasting, or thinness will be used based on WHO growth standards [19]. The nutritional indicators are labeled as stunting (height-for-age < -2 standard deviation [SD]), underweight/wasting (body mass index (BMI)-for-age < -2SD) [19, 20]. BMI-for age is used for adolescents from 10 to 19 years of age to assess adolescents' thinness or underweight. Therefore, all articles to be selected and included in this meta-analysis will be based on WHO growth standards parameters.

### Screening and selection process

The screening and selection process will be conducted through different inclusion and exclusion phases. In the first phase, all the relevant articles will be collected and the duplicates will

be identified and removed. In the second phase, the publications containing the search criteria in the title, in the keywords and in the abstract will be included. The full-text articles will be assessed following the above certain exclusion and inclusion criteria in the third phase. During this process, if the full text of the study is not available, the reviewers will contact the author to provide the full text. Nevertheless, if the authors do not respond or provide the full-text after two-mail contacts, then the article will be excluded from the review because of the inability to assess the quality of articles in the absence of full text. All the selected articles approved by both reviewers will be included in the study. If any discrepancy arises among the reviewers (J.P., L. J., and J.B.), it will be resolved through discussion and mutual consent, or else the next reviewer (S. K. A.) will be consulted if necessary. Articles relating to editorials, review articles, and case reports will be excluded from this meta-analysis.

## Data extraction

The reviewers (J.P. and S. K. A.) will screen the articles independently based on the title and abstract. Then relevant data will be extracted independently from the eligible articles by both the reviewers. Any differences in opinion are identified, it will be resolved through discussion (with a third reviewer where necessary). Missing data will be requested from the corresponding author(s) via e-mail. A predefined data extraction format on Microsoft Excel spreadsheet will be used to collect information on the name of the author/s, year of publication, study design, place of the study, region of the study, sample size, age range of study participants, sex of study participants, prevalence of stunting, prevalence of wasting or underweight and pertinent risk factors or determinants of undernutrition will also be extracted.

## Risk of bias and quality assessment

The reviewers (J.P., S. K. A., P. K. S.) will assess the quality of studies by adopting the Newcastle-Ottawa Quality Assessment Scale (NOS) for cross-sectional, case-control, and cohort studies [21]. The parameters to be used for quality assessment are representativeness of the sample (Sample strategy), age, sample size, cut-offs, and reference measures for adolescents under nutritional status. The final scoring system will be comprised of 10 criteria for rating the different quality elements for each study. This assessment scale will be used to assess the internal and external validity, risk of bias, and methodological quality of each included cross-sectional, cohort, and case-control study. The quality assessment tool has 3 sections. The first section will focus on the methodological quality of each original study such as objectives, sample size, and sampling technique. This section will be graded on the basis of 5 stars. The second section of the tool will consider the comparability of studies and will be graded out of 2 stars. The third section of the tool will consider the outcome measures and data analysis and will be graded out of 3 stars. Studies with $\geq$ 6 scores will be included in the review and meta-analysis of prevalence. Quality assessment will be checked independently by the four authors, and any disagreements will be resolved by discussion and mutual consensus. If the discrepancy still persists, the average scores of the reviewers will be taken into consideration. Similarly, in the case of determinants, each determinant or factor with the outcome variable will be critically assessed. Similar cut-off points will be employed for all prevalence studies of undernutrition. The level of risk of bias in each of the parameters will be presented separately for each study in tables in the final draft of the review.

## Data analysis and assessment of publication bias

This meta-analysis will be conducted using STATA V.16 statistical software. The extracted data from each study using a Microsoft Excel spreadsheet will be imported to this STATA

software. The prevalence of undernutrition (stunting, wasting/underweight) will be calculated by dividing the number of positive responses by the total study participants. The standard error of prevalence for each original article will be calculated using the binomial distribution formula. Undernutrition-specific estimates will be pooled separately for stunting and underweight/ wasting using a random-effects model [22]. The random-effect model at 95% CI will be used for estimating the overall effect. The effects of the selected associated factors on the outcome variable will be examined using separate groups of meta-analysis.

The existence of heterogeneity among studies will be assessed by computing p-values of Higgins's $I^2$ test statistics and Cochran's Q-statistics based on chi-square with a 5% level of significance among reported prevalence [22, 23]. Higgins's $I^2$ statistic measures the difference between sample estimates (in percentage) which is due to heterogeneity rather than to sampling error. The heterogeneity of the studies will be classified as low, moderate, and high based on the $I^2$ value less than 25%, 50%, and 75%, respectively [23].

Sensitivity analysis and subgroup analysis will be conducted based on study quality in order to investigate the possible sources of heterogeneity. The presence of publication bias will be assessed through visual examination of funnel plots [24] and statistically supported with confirmatory and/or objectivity testing with Egger's regression test at a 5% level of significance [25]. The funnel plot displays the studies included in meta-analysis in a plot of effect size against sample size [24]. A nonparametric trim and fill analysis [26] will be used to handle the observed publication bias for estimating the number of missing studies that might exist and in reducing and adjusting publication bias in meta-analysis.

## Ethics and dissemination

As this is a systematic review and meta-analysis, ethical approval is not required. The results of this review will be published in a peer-reviewed journal and will be presented at relevant conferences.

## Discussion

Adolescents are the most vulnerable population to undernutrition in India which becomes a major public health challenge for the country [10]. Dietary patterns and physical activity, socio-economic conditions, in addition to schooling (drop-outs rate), and prevailing social norms for early marriage influence the health and nutritional well-being of adolescents [27]. Undernutrition has numerous long-term consequences such as poor physical and mental function, increased vulnerability to infections, developing non-communicable diseases in adulthood, and economic burden for the healthcare system [28]. It is an urgent need to consider potential implications for policy and practice.

This planned review and meta-analysis will systematically explore the evidence available on the prevalence of undernutrition among Indian adolescents and also to identify the risk factors or determinants associated with undernutrition. By collecting and summarizing information can be a step towards better understanding of the prevalence of nutritional status of Indian adolescents and how the associated factors influence this prevalence. This review will provide directions for further research and healthcare practitioners.

## Supporting information

**S1 File. PRISMA-P checklist.**
(DOCX)

**S2 File. Draft of minimal dataset.**
(DOCX)

## Author Contributions

**Conceptualization:** Jayashree Parida, Subhendu Kumar Acharya.

**Investigation:** Jayashree Parida, Lopamudra Jena Samanta, Jagatdarshi Badamali, Subhendu Kumar Acharya.

**Methodology:** Jayashree Parida, Prasanna Kumar Patra, Bijay Kumar Mishra, Subhendu Kumar Acharya.

**Resources:** Jayashree Parida, Lopamudra Jena Samanta, Jagatdarshi Badamali.

**Visualization:** Jayashree Parida.

**Writing – original draft:** Jayashree Parida, Subhendu Kumar Acharya.

**Writing – review & editing:** Jayashree Parida, Prasant Kumar Singh, Sanghamitra Pati, Harpreet Kaur, Subhendu Kumar Acharya.

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
