## [Decision Letter · Decision Letter 0]

20 Dec 2021

PONE-D-21-34587Prevalence and Determinants of Undernutrition among Adolescents in India: A Protocol for Systematic Review and Meta-AnalysisPLOS ONE

Dear Dr. Acharya,

Thank you for submitting your manuscript to PLOS ONE. After careful consideration, we feel that it has merit but does not fully meet PLOS ONE’s publication criteria as it currently stands. Therefore, we invite you to submit a revised version of the manuscript that addresses the points raised during the review process.

Dear authors,

please follow the appointments done by the referees, and after, submit the altered manuscript, with answers point-to-point in a separate file.

Best Regards

Academic Editor

We look forward to receiving your revised manuscript.

Kind regards,

Eduardo Monguilhott Dalmarco, PhD

Academic Editor

PLOS ONE

Journal Requirements:

2. Please amend either the abstract on the online submission form (via Edit Submission) or the abstract in the manuscript so that they are identical.

Additional Editor Comments:

Dear authors,

please follow and improve the quality of the protocol submitted according to the referee's reports.

Reviewers' comments:

Reviewer's Responses to Questions

**Comments to the Author**

1. Does the manuscript provide a valid rationale for the proposed study, with clearly identified and justified research questions?

Reviewer #1: Yes

Reviewer #2: Yes

2. Is the protocol technically sound and planned in a manner that will lead to a meaningful outcome and allow testing the stated hypotheses?

Reviewer #1: Yes

Reviewer #2: Yes

3. Is the methodology feasible and described in sufficient detail to allow the work to be replicable?

Reviewer #1: No

Reviewer #2: Yes

4. Have the authors described where all data underlying the findings will be made available when the study is complete?

Reviewer #1: No

Reviewer #2: No

5. Is the manuscript presented in an intelligible fashion and written in standard English?

Reviewer #1: Yes

Reviewer #2: No

6. Review Comments to the Author

You may also provide optional suggestions and comments to authors that they might find helpful in planning their study.

Reviewer #1: PONE-D-21-34587 Parida, Jayashree et al. comments - Prevalence and Determinants of Undernutrition among Adolescents in India: A Protocol for Systematic Review and Meta-Analysis

The manuscript presents a protocol for a Systematic Review and Meta-Analysis that aims to determine the pooled prevalence and determinants of undernutrition (stunting, underweight/wasting) among Indian adolescents. The protocol was prepared according to (PRISMA-P 2009), Meta-analysis of Observational Studies in Epidemiology and PRISMA-P 2015 checklist. A systematic search in 6 databases (PubMed, EMBASE, SCOPUS, Google, Google Scholar, and Cochrane databases) was proposed and the quality of the articles included in the review will be evaluated using the Newcastle-Ottawa Scale (NOS). Publication bias will be assessed through visual examination of funnel plots and objectively by Egger’s regression test.

Major:

It would be insightful for authors to provide a more detailed, expanded and clarified transcript of a draft search strategy to be used for at least one electronic database (for e.g. Pubmed), including planned limits, to ensure that it could be repeated.

-The authors registered the review protocol at PROSPERO. Author´s should kindly provide PROSPERO’s data of registration and registration number.

Minor:

-According to PRISMA-P 2015 checklist, e-mail address of all protocol authors should be provided.

-Regarding PRISMA-P 2015 checklist item n° 7, an explicit statement of the question(s) the review will address with reference to participants, interventions, comparators, and outcomes (PICO) was not found on page number 3. The authors should kindly review the pages pointed at each checklist item of PRISMA-P 2015 checklist, in order to verify if the information requested matches the pages.

Reviewer #2: In this manuscript, the authors presented the “Prevalence and Determinants of Undernutrition among Adolescents in India: A Protocol for Systematic Review and Meta-Analysis”. The manuscript has an interesting topic to study through a systematic review and meta-analysis, the proposed study is clear and valid, the hypothesis is good and the methodology is feasible; however, some issues need to be addressed.

General

• Please, revise the English grammar, vocabulary, and punctuation. There are grammatical errors and sometimes it is hard to understand the text.

Abstract

• Methodology: You should include the registration number of the PROSPERO protocol.

• Discussion: in the last sentence you could include the contribution to the creation or improvement of public policies to fight undernutrition.

Introduction

• In the 2nd paragrapher you should provide an explicit statement of the question(s) the review will address with reference to participants, interventions, comparators, and outcomes (PICO) (according to the PRISMA)

Objectives

• You might review the format of the objective.

Methods

Systematic review registration and reporting of the findings

• You should review the PRISMA-P checklist (attached file);

• You have to include the PROSPERO number.

Study design and search strategy

• Did you do a pilot search to define the MeSH terms and search strategies? Have you validated them?

• You should include more keywords as synonyms to adolescents like teenagers or young adults.

• You should organize the MeSH terms into blocks to retrieve the publications and show them.

Inclusion criteria

• “All articles published only in the English language” Why not in Hindi?

Risk of bias and quality assessment

• Why do not submit the studies to a critical appraisal using the Joanna Briggs Institute Critical Appraisal checklist (JBI 2020) and Grading of Recommendations Assessment, Development, and Evaluation (GRADE)?

Data analysis and assessment of publication bias

• How many studies do you need to perform the meta-analysis?

7. PLOS authors have the option to publish the peer review history of their article (what does this mean?). If published, this will include your full peer review and any attached files.

Reviewer #1: No

Reviewer #2: No

---

## [Author Response · Author response to Decision Letter 0]

31 Dec 2021

Rebuttal letter

We thank the academic editor and the reviewers for their reviews and valuable comments on our manuscript entitled “Prevalence and Determinants of Undernutrition among Adolescents in India: A Protocol for Systematic Review and Meta-Analysis”. We have now revised the manuscript as per the comments and suggestions. 

We have included a copy of the manuscript with track changes labelled “Revised Manuscript with Track Changes” and a revised manuscript without track changes labelled “Revised Manuscript”. We have also provided a point-by-point responses to the comments made by the academic editor and reviewers.

We hope the responses will be satisfactory for the academic editor and the reviewers for further consideration for publication. 

With warm regards, 

Dr. Subhendu Acharya

On behalf of all authors,

Academic Editor’s Comments:

Response: File naming was edited to comply with the style requirements. We hopefully have no divergences from the style requirements now.

1. Please amend either the abstract on the online submission form (via Edit Submission) or the abstract in the manuscript so that they are identical.

Response: The abstract was corrected in the manuscript as well as on the online submission form.

Response: This has been deleted from all sections other than the methods.

Additional Editor Comments:

Dear authors,

Please follow and improve the quality of the protocol submitted according to the referee's reports.

Response: We have revised and addressed the points according to the reviewers’ report. 

Reviewers' comments:

Reviewer #1

PONE-D-21-34587 Parida, Jayashree et al. comments - Prevalence and Determinants of Undernutrition among Adolescents in India: A Protocol for Systematic Review and Meta-Analysis

The manuscript presents a protocol for a Systematic Review and Meta-Analysis that aims to determine the pooled prevalence and determinants of undernutrition (stunting, underweight/wasting) among Indian adolescents. The protocol was prepared according to (PRISMA-P 2009), Meta-analysis of Observational Studies in Epidemiology and PRISMA-P 2015 checklist. A systematic search in 6 databases (PubMed, EMBASE, SCOPUS, Google, Google Scholar, and Cochrane databases) was proposed and the quality of the articles included in the review will be evaluated using the Newcastle-Ottawa Scale (NOS). Publication bias will be assessed through visual examination of funnel plots and objectively by Egger’s regression test.

Major:

1. It would be insightful for authors to provide a more detailed, expanded and clarified transcript of a draft search strategy to be used for at least one electronic database (for e.g., Pubmed), including planned limits, to ensure that it could be repeated.

Response: We added search strategy of one electronic database i.e., Pub Med

2. The authors registered the review protocol at PROSPERO. Author´s should kindly provide PROSPERO’s data of registration and registration number.

Response: PROSPERO registration number was added.

Minor:

1. According to PRISMA-P 2015 checklist, e-mail address of all protocol authors should be provided.

Response: We corrected and added e-mail address of all protocol authors

2. Regarding PRISMA-P 2015 checklist item n° 7, an explicit statement of the question(s) the review will address with reference to participants, interventions, comparators, and outcomes (PICO) was not found on page number 3. The authors should kindly review the pages pointed at each checklist item of PRISMA-P 2015 checklist, in order to verify if the information requested matches the pages.

Response: We corrected the page numbers accordingly. 

Reviewer #2

In this manuscript, the authors presented the “Prevalence and Determinants of Undernutrition among Adolescents in India: A Protocol for Systematic Review and Meta-Analysis”. The manuscript has an interesting topic to study through a systematic review and meta-analysis, the proposed study is clear and valid, the hypothesis is good and the methodology is feasible; however, some issues need to be addressed.

General

1. Please, revise the English grammar, vocabulary, and punctuation. There are grammatical errors and sometimes it is hard to understand the text.

Response: Thank you for your valuable inputs. All the grammatical and typological errors were corrected at revision.

Abstract

2. Methodology: You should include the registration number of the PROSPERO protocol.

Response: PROSPERO registration number was added.

3. Discussion: in the last sentence you could include the contribution to the creation or improvement of public policies to fight undernutrition.

Response: Thank you for your comment. We added to the discussion part of the abstract section.

Introduction

4. In the 2nd paragraph you should provide an explicit statement of the question(s) the review will address with reference to participants, interventions, comparators, and outcomes (PICO) (according to the PRISMA)

Response: We added research questions in the 2nd paragraph of introduction section, before objectives.

Objectives

5. You might review the format of the objective.

Response: It was revised.

Methods

Systematic review registration and reporting of the findings

6. You should review the PRISMA-P checklist (attached file)

Response: We corrected it.

7. You have to include the PROSPERO number.

Response: We added the PROSPERO registration number

Study design and search strategy

8. Did you do a pilot search to define the MeSH terms and search strategies? Have you validated them?

Response: Yes, we did a pilot search to define MeSH terms and also search strategies. We constructed gold standard reference set to validate the MeSH terms and search strategies. 

9. You should include more keywords as synonyms to adolescents like teenagers or young adults.

Response: Thank you for your suggestion. We added other keywords like teenagers or young adults. 

10. You should organize the MeSH terms into blocks to retrieve the publications and show them.

Response: We have included one table (supplementary file-S1) on Mesh terms in the search strategy section of revised manuscript.

Inclusion criteria

11. “All articles published only in the English language” Why not in Hindi?

Response: It is time consuming and costly to translate the non-English language papers. Searching for Indian languages using word formation techniques is not available.

Risk of bias and quality assessment

12. Why do not submit the studies to a critical appraisal using the Joanna Briggs Institute Critical Appraisal checklist (JBI 2020) and Grading of Recommendations Assessment, Development, and Evaluation (GRADE)?

Response: The Newcastle-Ottawa Scale (NOS) is one of the most commonly used tools worldwide for evaluating quality in meta-analysis of observational studies. It is possible to use it as potential moderator in meta-regression analyses (Veronese et al. 2016).

Data analysis and assessment of publication bias

13. How many studies do you need to perform the meta-analysis?

Response: Theoretically, there is no such restriction for number of studies in meta-analysis. we can perform meta-analysis with just two studies. However, more studies means that meta-analysis have more power and is more exact and reliable.

---

## [Editor Report · Decision Letter 1]

11 Jan 2022

Prevalence and Determinants of Undernutrition among Adolescents in India: A Protocol for Systematic Review and Meta-Analysis

PONE-D-21-34587R1

Dear Dr. Acharya,

We’re pleased to inform you that your manuscript has been judged scientifically suitable for publication and will be formally accepted for publication once it meets all outstanding technical requirements.

Kind regards,

Eduardo Monguilhott Dalmarco, PhD

Academic Editor

PLOS ONE

Additional Editor Comments (optional):

The authors now as an academic editor, I consider that all questions raised by the referees were answered by the authors.
---

## [Editor Report · Acceptance letter]

13 Jan 2022

PONE-D-21-34587R1 

Prevalence and Determinants of Undernutrition among Adolescents in India: A Protocol for Systematic Review and Meta-Analysis 

Dear Dr. Acharya:

I'm pleased to inform you that your manuscript has been deemed suitable for publication in PLOS ONE. Congratulations! Your manuscript is now with our production department. 

Kind regards, 

on behalf of

Professor Eduardo Monguilhott Dalmarco 

Academic Editor

PLOS ONE